# Assessing Animal Models to Study Impaired and Chronic Wounds

**DOI:** 10.3390/ijms25073837

**Published:** 2024-03-29

**Authors:** Shayan Saeed, Manuela Martins-Green

**Affiliations:** Department of Molecular, Cell, and Systems Biology, University of California, Riverside, CA 92521, USA; ssaee002@ucr.edu

**Keywords:** wound healing, pre-clinical models, diabetes, rodent, porcine, chronic wounds, impaired healing wounds

## Abstract

Impaired healing wounds do not proceed through the normal healing processes in a timely and orderly manner, and while they do eventually heal, their healing is not optimal. Chronic wounds, on the other hand, remain unhealed for weeks or months. In the US alone, chronic wounds impact ~8.5 million people and cost ~USD 28–90 billion per year, not accounting for the psychological and physical pain and emotional suffering that patients endure. These numbers are only expected to rise in the future as the elderly populations and the incidence of comorbidities such as diabetes, hypertension, and obesity increase. Over the last few decades, scientists have used a variety of approaches to treat chronic wounds, but unfortunately, to date, there is no effective treatment. Indeed, while there are thousands of drugs to combat cancer, there is only one single drug approved for the treatment of chronic wounds. This is in part because wound healing is a very complex process involving many phases that must occur sequentially and in a timely manner. Furthermore, models that fully mimic human chronic wounds have not been developed. In this review, we assess various models currently being used to study the biology of impaired healing and chronic non-healing wounds. Among them, this paper also highlights one model which shows significant promise; this model uses aged and obese *db*/*db*^−/−^ mice and the chronic wounds that develop show characteristics of human chronic wounds that include increased oxidative stress, chronic inflammation, damaged microvasculature, abnormal collagen matrix deposition, a lack of re-epithelialization, and the spontaneous development of multi-bacterial biofilm. We also discuss how important it is that we continue to develop chronic wound models that more closely mimic those of humans and that can be used to test potential treatments to heal chronic wounds.

## 1. Introduction

Impaired wounds are wounds that fail to proceed through the normal acute healing processes in a timely and orderly manner but eventually heal, albeit not optimally. Chronic wounds, on the other hand, are wounds that remain unhealed for weeks and months. According to a recent consensus panel of expert physicians and research scientists, wounds can be considered chronic if they fail to progress towards healing within 4 weeks following the initiation of standard treatment [1]. Chronic wounds occur in people with systemic physiological problems such as diabetes mellitus, aging, obesity, and peripheral vascular disease, and/or who are exposed to external factors such as microbial infections and environmental pollutants. Although their exact etiology remains unknown, they are characterized by elevated levels of reactive oxygen species (ROS), chronic inflammation, lack of blood circulation, high levels of pro-inflammatory cytokines and proteases, as well as infection with biofilm-forming microbes [2].

It has been reported that 1–2% of people in developed countries will have chronic wounds at least once in their lifetime. In the US alone, they impact ~8.5 million people and cost ~USD 28–90 billion/year, not accounting for the pain and suffering that patients endure, both psychologically and physically. Many treatment protocols have been developed throughout the last few decades. For example, wounds are debrided, infection and pain are controlled, standard treatments are performed, and dressings are applied. Moreover, grafts, hyperbaric oxygen treatment, and many cellular and tissue products designed to mimic the properties of normal skin have been developed to treat chronic wounds, but so far, none have been truly effective [3]. Therefore, more research needs to be carried out to understand how wounds become chronic so that new approaches can be developed to switch the path to wound chronicity into the path of wound healing. This is particularly important when wounds are debrided; an optimal outcome would be that the wound can now heal as opposed to remaining chronic. This review will primarily focus on preclinical animal models of impaired healing and diabetic chronic wounds, rather than alternative in vitro models. Although in vitro skin equivalents have been made in recent years that almost completely mimic human skin morphology and physiology, models that accurately replicate wound-healing pathologies, such as chronic wounds, are still in their infancy and require major advancements before they can be used in studies of chronic wound etiology and to test novel therapies [4,5,6,7,8,9,10].

## 2. Models to Study Impaired Wound Healing

Impaired healing occurs primarily in aged and malnourished individuals, as well as people with cardiovascular disease. Several models have been generated that show impaired wound healing. Researchers developed a mouse tail model to study delayed acute wound healing in normal mice. This model involves creating 1 × 0.3 cm full-thickness wounds at the base of the tail [11]. Tail wounds require 14 to 25 days for complete wound closure as opposed to the 10–12 days required by full-thickness dorsal wounds [11,12],. This provides researchers with a longer timeframe to study mechanisms of delayed healing and to test new drugs and/or therapies that improve the healing processes. However, given the differences in anatomy between the tail and dorsum, understanding how this model can provide insights into the impaired healing of cutaneous wounds is difficult. Nonetheless, this model may be useful in studies of wound re-epithelialization.

Wounds in diabetic individuals are characterized by excessive oxidative stress, increased inflammation with the excessive presence of pro-inflammatory macrophages (M1), a paucity of anti-inflammatory macrophages (M2), decreased angiogenesis, and a lack of granulation tissue formation [13]. These conditions predispose many diabetic patients to infections, further hindering the healing process. In addition, other complications include tissue hypoxia due to the associated peripheral vascular disease, peripheral neuropathy, and hyperglycemia, known to increase cell apoptosis [14]. Several studies have generated models, as described below, reflecting the pathophysiology of diabetes, which can be used to study impaired healing associated with diabetes.

### 2.1. Chemically Induced T1DM Models

Impaired healing occurs frequently in people who suffer from diabetes mellitus (DM), both Type 1 (T1DM) and Type 2 (T2DM). T1DM has historically been induced in either mice or rats that have been injected with alloxan or streptozotocin (STZ), both of which selectively attack β cells of the islets in the pancreas, resulting in insulin deficiency and hyperglycemia. However, the use of STZ has been preferred over alloxan because of the longer and relatively more stable induced- hyperglycemia by this drug, less associated toxicity, and well-characterized diabetic complications [15,16]. Moreover, although STZ alone generates a model exclusively of T1DM, some strategies have been developed that also allow T2DM to be modeled [15,17]. The first involves the partial protection of β cells of the islets in the pancreas by concurrently administrating a compound that protects against the diabetogenic effect of STZ. Although this method creates insulin deficiency and moderate levels of hyperglycemia, it does not result in resistance to insulin, an important characteristic of T2DM. The second strategy involves exposing the animal to a high-fat diet to create resistance to insulin, followed by the administration of STZ to reduce β-cell capacity. This produces hyperglycemia along with hyperinsulinemia and insulin resistance, better resembling human T2DM [15]. However, the characterization of wound healing in the T2DM models generated in this manner has not been reported.

In general, two protocols are used for the generation of T1DM with STZ treatment: a single large dose of STZ or multiple administrations of low-dose STZ over several days. A single large dose (200 mg/kg) causes the toxin-induced necrosis of pancreatic islet β cells and results in diabetes within 48 h, with blood glucose concentrations measuring >500 mg/dL. However, this model lacks some important features of T1DM, most notably pancreatic insulitis [15]. On the other hand, multiple administrations of low doses show partial damage to pancreatic islet cells and an inflammatory response that further damages β cells, resulting in hyperglycemia alongside lymphocyte infiltration in the pancreatic islets, insulitis, and insulin deficiency [15,18]. This method shows better resemblance to human T1DM and has fewer toxic effects than a single dose. For this reason, it has become increasingly popular in recent studies. But despite this, findings from experiments that employ STZ models of DM should be cautiously extrapolated to humans given that STZ is also a toxin to organs and tissues other than just pancreatic islet cells [15]. Figure 1A summarizes both protocols. 

The use of STZ for T1DM generation can be conducted in mice, rats, or pigs to study specific pathways in diabetic wound healing. A significant advantage of STZ-induced diabetes in mice is that it can be used in combination with genetically modified animals to study the roles of key genes in the wound-healing process. For example, studies investigating the role of Toll-like receptor-2 (TLR2) have been performed by treating TLR2^−/−^ mice with STZ [19]. Once T1DM was established (blood glucose levels exceeding 250 mg/dL), wounds were made on the dorsa of the mice [19]. These wounds showed decreased oxidative stress, NF-κB activation, and cytokine secretion, and had improved wound closure. Similar methods were used to induce diabetes in *RhoB*^+/−^ and *RhoB*^−/−^ [20], MMP9^−/−^ [21], and leukotriene-deficient 5LO^−/−^ mice [22] to study the role of these proteins in T1DM. 

The development of a diabetic impaired healing model in pigs could be of significant translational relevance for the study of human diabetic wounds given several important factors. As opposed to rodents, pig skin’s anatomy and physiology more closely resemble those of human skin. Additionally, their larger size permits more wounds, both control and experimental, on the same animal, allowing for greater statistical significance [23]. Pig models of diabetic wounds involved inducing diabetes by administering STZ at a dose of 150 mg/kg in Yorkshire pigs, as illustrated in Figure 1B. Persistent hyperglycemia with blood glucose levels greater than 350 mg/dL was established within the first 24 h, and full-thickness excision wounds were made on the dorsum, 2 weeks after STZ treatment. Re-epithelialization was significantly delayed, and collected wound fluid had marked reductions in the levels of IGF-1 and TGF-β, but not of PDGF, compared to non-diabetic pigs [24]. Human human diabetic wounds also have low levels of IGF-1 [25]. Moreover, given that diabetes induced within just 2 days was sufficient to cause impaired healing, it was suggested that long-term complications from diabetes, such as neuropathy and vasculopathy, may not be responsible for the impaired healing of diabetic wounds [24]. 

Another pig model has also been investigated for bacterial infections using *Staphylococcus aureus* in a wound chamber [26]. The diabetic wounds showed sustained significant infection and delayed re-epithelialization compared to non-inoculated diabetic and non-diabetic wounds. This model has also been modified for studying burn wounds [27]. Partial-thickness burn wounds were generated on diabetic pigs three weeks after STZ injections, and delayed wound re-epithelialization was observed in diabetic pigs on days 7, 10, and 14 post-burn, but complete re-epithelialization occurred by day 21 across all animals. It was also noted that the success of STZ-induced hyperglycemia was dependent on the size of the pigs, with larger pigs (45–50 kg) presenting more consistent hyperglycemia than smaller pigs (25–30 kg) [27]. A significant limitation of this burn wound model was the inconsistency of the relative burn depths, because three weeks after the induction of diabetes and immediately before burn injury, the skin of diabetic pigs was found to be significantly thinner than normal pigs (a difference of 1.6 mm) and had less subcutaneous fat. Therefore, it was unclear whether the relative greater burn depth or systemic alterations due to a diabetic state was the cause of impaired healing [27]. Moreover, more comprehensive studies have shown that a slow injection of STZ at 130 mg/kg in pigs, combined with a moderate high-fat diet, results in a model of T2DM [28], suggesting that this model might be more suitable for studying burns in diabetes. 

Another group of researchers sought to establish a standardized diabetic pig model [29]. For these studies, Yorkshire pigs of 2–3 months of age weighing 20–25 kg were administered 150 mg/kg of STZ over 15–20 min, and blood glucose was maintained between 250 and 450 mg/dL during the course of the experiment by controlling food intake. Nine to twelve full-thickness wounds which were 1.5 cm × 1.5 cm were created on the dorsum, covered with clear bandages, and taped with cotton gauze cloth. The entire wounded area was covered with elastic bandages and wrapped in Elastikon. Although a previous study showed a significant delay in healing of 4 to 6 days in pigs wounded 14–20 days after diabetes induction [24], the authors of this study did not detect such a delay. Instead, the authors conducted a series of tests whereby wounding was undertaken 20, 45, and 90 days after STZ injection and observed significant delays in wound closure in pigs wounded 45 days after STZ injections. In these pigs, wounds remained open on day 14 and closed on day 21. Delays were even more pronounced in pigs injured 90 days after STZ injection, with wounds remaining open even on day 21. This data suggests that the longer the condition of diabetes is maintained in pigs prior to wounding, the more pronounced the delay in healing [27]. Therefore, it can be concluded that exposure to prolonged hyperglycemic conditions is required to study the effects of diabetes on wound healing.

### 2.2. Genetically Modified T2DM

Genetically modified strains of mice have become increasingly popular to model wound healing of T2DM. Currently, the most widely used strains include the *ob*/*ob* (with mutations in the leptin gene) and the *db*/*db* strains (with mutations in the leptin receptor gene), both characterized by excessive feeding (hyperphagia) resulting in obesity and hyperglycemia. Furthermore, the genetic background of the strain plays an important role in the phenotypic outcome. In the C57BLKS/J genetic background, both *ob*/*ob* and *db*/*db* mutant mice manifest morbid obesity, chronic hyperglycemia, and pancreatic β cell atrophy. In contrast, in the C57BL/6J genetic background, *ob*/*ob* and *db*/*db* mutant mice manifest only transient hyperglycemia, and β cells show hypertrophy, not atrophy [30]. It has recently been found that the excessive adipose tissue present in these strains causes macrophage activation that plays a key role in insulin resistance [31]. In addition to the *ob*/*ob* and *db*/*db* strains, newer strains, including NONcNZO10/LtJ and TALLYHO, have also been recently shown to be possible models for impaired wound healing studies. Both show early β cell hypertrophy followed by atrophy in the later stages of diabetes as well as elevated levels of triglycerides. In addition, their moderate obesity and delayed hyperglycemia may more accurately reflect the effects of diabetes on some humans with T2DM [30].

Goodson and Hunt (1986) were among the earliest investigators to characterize wound-healing complications in a diabetic mouse model [32]. Full-thickness 6 mm in diameter excision wounds were generated on 8–10-week-old C57BL-*ob*/*ob* mice and either an expended polytetrafluoroethylene (ePTFE) sponge or polyvinyl collagen alcohol sponge was implanted during 10–14 days for collagen characterization. The results showed significant collagen deficiency and delayed wound closure in the *ob*/*ob* mice. However, early insulin treatment and dietary restriction in some *ob*/*ob* mice resulted in collagen accumulation and glucose levels similar to those of control mice [32]. These results are likely due to the short duration of exposure of the mouse body to hyperglycemia, being just 8–10 weeks old at the time of wounding. As shown previously, the duration of hyperglycemia exposure is important for the degree of effects on wound healing [29].

In the *db*/*db* mice, a model of impaired healing was reported whereby full-thickness excision wounds of 1.5 cm × 1.5 cm were performed on C57BL/KsJ-*db/db^−/−^
* followed by the placement of a polyurethane dressing [33]. The investigators observed a significant decrease in inflammatory cells in the wound tissue as well as decreased granulation tissue formation and a slower rate of wound closure (4–6 weeks as opposed to 10–16 days for control). They also found that the topical application of either *rPDGF*-*BB* or *rbFGF* resulted in increased fibroblast proliferation, improved capillary formation, and wound closure [33]. Other investigators increased wound-healing impairment in a *db/db^−/−^
* mouse model by placing silicon splints around full-thickness excision wounds to prevent wound contraction [34]. When compared to stented wounds generated on other genetically derived diabetic models (e.g., Akita, replicating T1DM) and STZ-induced models (also T1DM), the *db/db^−/−^
* strain showed significant delays in wound closure, decreased granulation tissue formation, and wound bed vascularization [34]. 

Although the *ob*/*ob* and *db*/*db* mice strains have become popular for studies of impaired wound healing, they must be used with caution given that a single mutation in the leptin or leptin receptor gene cannot exhaustively replicate the complex pathophysiology of diabetic impaired wounds. Indeed, in the uncommon occurrence of a leptin receptor mutation in humans, studies have indicated that its role in humans is significantly more complex than in mice [35]. 

Considering the limitations of both the *ob*/*ob* and *db*/*db* mice strains, other strains have been recently developed to more accurately represent human diabetes. A monogenic *Akita* mouse strain (representative of T1DM because of a mutation in insulin gene that inhibits secretion) and a polygenic mouse strain, NONcNZO10/LtJ (representative of T2DM), both of which lack many of the extreme symptoms of leptin or leptin receptor gene mutation strains, were compared with STZ-induced and *db*/*db* mice for their potential for impaired wound healing [36]. Generated incisional, splinted excision wounds and ischemia–reperfusion injury failed to show any significant wound-healing deficits in the *Akita* strain. While the *db*/*db* strain showed wound-healing deficiencies in excision and incision wounds, the NONcNZO10 strain showed healing impairment in all three types of wounds: incision, splinted excision, and ischemia–reperfusion wounds [36]. The administration of FGF-1 to splinted excision wounds made in NONcNZO10 mice showed improved re-epithelialization [37]. Investigators have also used a TallyHo/JnJ diabetic mouse strain to create a model of incision, splinted excision, and ischemia–reperfusion wounds [38]. All three models showed decreased angiogenesis, collagen formation, re-epithelialization, granulation tissue formation as well as the downregulation of many important cytokines [38], but none showed wound chronicity (i.e., biofilm and prolonged non-healing). Table 1 summarizes models of chemically and genetically induced models of impaired wounds in both murine and porcine.

## 3. Models to Study Chronic Wounds

### 3.1. Murine Models

It is well known that high levels of oxidative stress (OS) are present in human chronic wounds [39,40]. Under this premise, a mouse model of chronic wounds was developed by Kim et al. [41] in *db/db^−/−^
* mice that mimics chronic wounds in humans by increasing the levels of ROS in the wound tissue [41,42,43]. To achieve that goal, two major antioxidant enzymes were inhibited: catalase by the IP injection of 3-amino-1,2,4-triazole (ATZ) 20 min prior to wounding and Glutathione Peroxidase (GPx) by the topical administration of mercaptosuccinic acid (MSA) immediately after wounding (see Figure 2 for protocol summary). This treatment was performed only once at the time of wounding. This model accounts for aging by using *db/db^−/−^
* mice aged 5–6 months, as opposed to the commonly used 2-month-old mice, to ensure they have experienced prolonged hyperglycemia and fully developed diabetes and obesity [27,43]. Unlike previous diabetically impaired wound models, this model forms chronic ulcers for many weeks (some for months if the mouse does not die). These ulcers have increased oxidative stress, chronic inflammation, impaired dermal–epidermal connectivity, damaged microvasculature (e.g., fibrin cuffs), an abnormal collagen matrix in the wound tissue, a lack of re-epithelialization, and the spontaneous development of multi-bacterial biofilm, all characteristics of chronic wounds in humans. Treatment with the antioxidant agents N-acetyl cysteine (NAC) and α-tocopherol (α-toc) showed a reduction in oxidative stress, improvement in granulation tissue formation, sensitivity of the bacteria to antibiotic treatment, and wound closure [42]. Using this model, it was also shown that there is a dose-dependent effect of OS levels on wound chronicity [41]. The establishment of varying OS levels was accomplished by administering different concentrations of ATZ and MSA at wounding. Moreover, it was also shown that wound chronicity required an increase in both OS and bacteria from the *db/db^−/−^
* mouse skin microbiome, highlighting their critical role in the development of chronic wounds [43]. Thus, these findings showed that both elevated levels of OS and the presence of skin bacteria are critical for chronic wound development. More recently, sequencing of the bacterial internal transcribed spacer (ITS) gene of the wound microbiome from the chronic wounds in this mouse model was conducted at various time points post-wounding until 20 days post-wounding, when the wounds were fully chronic. Whereas non-chronic wounds (wounds not treated with MSA or ATZ) had a diverse microbiota similar to that of normal skin and went on to heal, in chronic wounds, the microbiome was much less diverse, the biofilm-forming bacteria predominated, and the wounds did not heal [44].

Other investigators have attempted to replicate the above model in an STZ-induced diabetic mouse model [45] (see Figure 3A). Mice were treated with MSA and ATZ following wounding, but *S. aureus* had to be inoculated, unlike in the genetically modified model of chronic wounds described above, in which the wounds were infected by *Staphylococcus* spontaneously from the microbiome of the skin [44]. Additionally, at the time of wound tissue collection, wounds looked more as if covered by a heavy dry scab as opposed to being infected and there was no reporting on the duration of ulceration [45].

Another group recently used *db/db^−/−^
* mice and created chronic wounds in 11-week-old mice (as opposed to 6-month-old mice) [46]. They used the same approach as Dhall et al. [42] and Kim et al. [41] but altered the dosage of the drugs used to create high oxidative stress. The mice were injected with half the dosage of ATZ (0.5 g/kg body weight compared to 1 g/kg body weight) and double the dosage of MSA was topically applied (300 mg/kg body weight compared to 150 mg/kg body weight), but MSA was washed off with saline five minutes after application (see Figure 3B). Whereas the ATZ- and MSA-treated mice showed delayed wound contraction, higher levels of inflammation, collagen deposition, cellular proliferation, and keratinocyte and leukocyte infiltration, wounds failed to show significant biofilm formation and delayed healing when compared to the control, a key characteristic of human chronic wounds [46].

In addition to these more recent models, another model was developed in 2010 that attempted to replicate biofilm-infected diabetic wounds [47]. Full-thickness excision wounds generated on the dorsum of *db/db^−/−^
* mice were inoculated with *Pseudomonas aeruginosa* biofilm 48hrs after the initial wounding procedure, and then, the wounds were covered with semi-occlusive dressings (see Figure 3C). Unlike the control *db*/*db*^−/−^ mice, wounds infected with these bacteria failed to heal in 28 days. Histological studies showed a significant inflammatory response, extensive epidermal hyperplasia, and tissue necrosis and most of the *Pseudomonas aeruginosa* that was inoculated was found in the scab and not in the wound tissue [47].

Yet another group in 2013 used a TallyHo (TH) mouse strain in which excision wounds were stabilized by silicon splints and *S. aureus* was immediately applied followed by a dressing [48]. In comparison to their background mice, the TH mice showed significantly less cytokine expression, including TLR 2, TLR 4, interleukin-1β, and tumor necrosis factor-α. Whereas similar neutrophil infiltration was observed in both groups, the neutrophils in the TH mouse showed reduced oxidative burst activity and significantly delayed wound re-epithelization 10 days after wounding [48]. However, the study lasted just 10 days post-wounding, and the model did not display signs of chronic wound development.

### 3.2. Porcine Models

Pig models may be the most suitable preclinical wound models given the similarities of their immune system and skin anatomy to those of humans [49,50]. If a model of chronic wounds can be created in pigs, such a model would be more clinically relevant. The first attempt to develop a chronic wound in pigs was performed by creating a burn wound [51]. Full-thickness burn injuries were generated on Yorkshire pigs followed by the mixed-species inoculation of *Acinetobacter baumannii* and *Pseudomonas aeruginosa* 3 days after wounding (see Figure 4). Previously developed [52] biofilm infection criteria was used to assess biofilm establishment in this pig model by supporting all three of its conditions: (1) *adherence to surface* tested by a scrubbing technique known to remove planktonic bacteria from burn wounds; (2) *persistent and localized infection* indicated by bacterial presence from day 14 to 35 following inoculation; and (3) *resistance to anti-microbial treatments* resulting from the lack of effectiveness of Acticoat 7™, a standard anti-bacterial dressing, in dismantling the biofilm and killing the bacteria. The wound biofilm also exhibited the elevated expression of genes involved in biofilm formation such as *rhlR, rpoS,* and *arpR* and, in addition, showed resilience to the debridement of necrotic and infected tissue [51]. Interestingly, although visual assessment showed that wound closure remained unaffected in wounds with biofilm, there was a significant reduction in skin barrier function, as assessed via transepidermal water loss, in comparison to non-inoculated wounds. These investigators also showed that the silencing of two tight-junction proteins, ZO-1 and ZO-2, by inducing miR-146a and miR-106b, was responsible for the resulting transepidermal water loss [51]. More recently, a similar method was used for the generation of chronic biofilm-infected wounds in Ossabaw pigs with metabolic syndrome that was induced via a high-fat diet [53]. The results showed impaired vascularization, reduced collagen levels, a lower abundance of endothelial cells, the reduced length of rete ridge (indicating diminished mechanical strength), and wound infection that persisted for at least 31 days. However, chronic wounds did not develop [53].

A chronic ischemic wound model in pigs that remained open for at least four weeks has also been developed [54]. Full-thickness bipedicle skin flaps measuring 15 × 5 cm were generated in these pigs and elevated by the placement of silicon sheets between the flap and subcutaneous tissue to prevent reperfusion. Full-thickness excision wounds were then created in the center of each flap. Doppler imaging showed that lower blood flow in the flaps and skin perfusion pressure, an indicator of microcirculation, were significantly compromised. Histological analysis showed impaired re-epithelization, delayed macrophage recruitment, decreased endothelial cell populations, and poorer organization. Wound tissue transcriptome analysis using a porcine GeneChip at various timepoints post-wounding showed the increased expression of SOD2 and arginase 1, both well-known markers of tissue oxidative stress that are also present in human chronic wounds. Given these phenotypic similarities with those of human chronic wounds, the authors have suggested that this model is a significant pre-clinical model for human chronic ischemic wounds [54]. However, this model is more appropriate to study processes involved in wound ischemia. Table 2 summarizes models of non-healing chronic wounds in both murine and porcine.

## 4. Conclusions

Although several models to study impaired and chronic wounds are available, the field continues to require models that can more closely mimic these diseases in humans and function as platforms for preclinical evaluation of potential therapies. In the case of chronic wounds, new models should take into consideration key characteristics of chronic wounds, such as infection and/or biofilm formation. If such goals are achieved, more effective treatments can potentially be developed. While previous treatments have used single-molecule strategies (e.g., PDGFBB) or are generalized (e.g., biological dressings), it is important that forthcoming treatments involve more complex approaches that can be used in consonance with the different phases of wound healing. Moreover, given that the environment of chronic wounds is severely damaging to biological molecules and cells, using small, stable molecules that can survive these harsh environments and are specific to treating biofilm without adversely affecting the host tissue is a potentially viable treatment for chronic wounds following debridement. At the same time, treatments should be affordable and available off the shelf to lower healthcare costs.

## Figures and Tables

**Figure 1 ijms-25-03837-f001:**
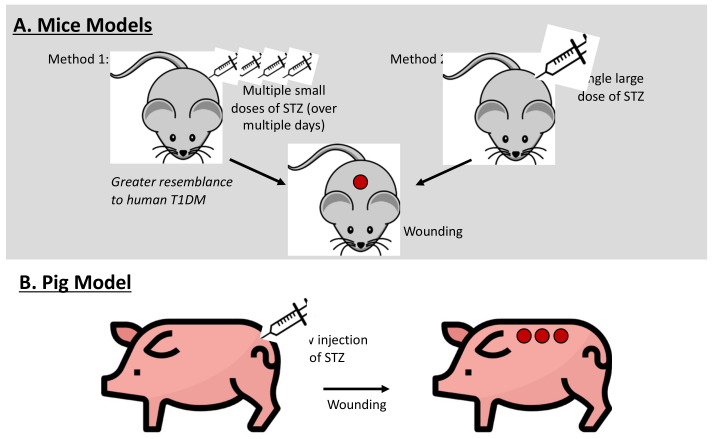
**Models of chemically induced impaired healing wounds:** (**A**) Mouse models for type 1 diabetes mellitus (T1DM) (1) by multiple small dosages of streptozotocin (STZ) over several days or (2) by a single large dosage of STZ followed by full-thickness excisional wounds on dorsum. STZ selectively attacks β cells of the islets in the pancreas, resulting in insulin deficiency and hyperglycemia; multiple small dosages of STZ in mice show better resemblance to human T1DM. (**B**) Similar STZ-induced diabetes mellitus models have been created in pig; slow injection of STZ combined with moderately high-fat diet may result in a representative model of T2DM. Studies seem to show strong correlation between length of diabetic state prior to wounding and delay in wound healing.

**Figure 2 ijms-25-03837-f002:**
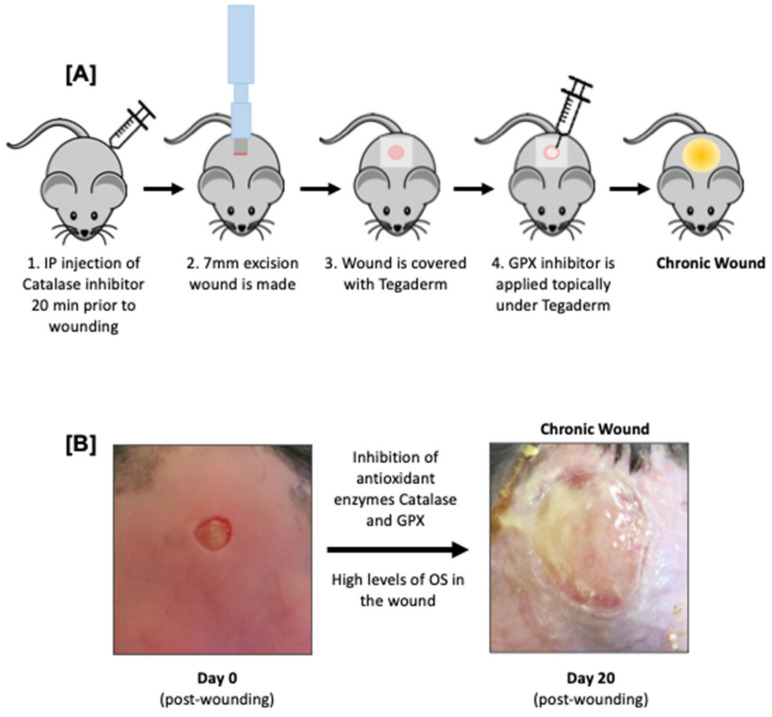
**Chronic wound model in *db/db*^−/−^ mice with high oxidative stress:** (**A**) The 5–6-month-old *db*/*db*^−/−^ mice receive (1) intraperitoneal (IP) injection of ATZ, a catalase inhibitor, 20 min prior to wounding. (2) Full-thickness 7 mm excisional wounds are created on the mouse dorsum via skin punch biopsy and scissors and (3) covered with Tegaderm. (4) MSA, a GPx inhibitor, is topically applied under the Tegaderm immediately after wounding. Biofilm naturally spawns within 3–5 days after wounding; 10–15 days after wounding, biofilm greatly increases and wound remains chronically opened and unhealed. (**B**) Images of wounds at Day 0 and Day 20 after injury, with the latter spontaneously harboring multi-bacterial biofilm and showing remarkably similar characteristics to human chronic wounds, including high oxidative stress, chronic inflammation, damaged microvasculature, and abnormal collagen matrix.

**Figure 3 ijms-25-03837-f003:**
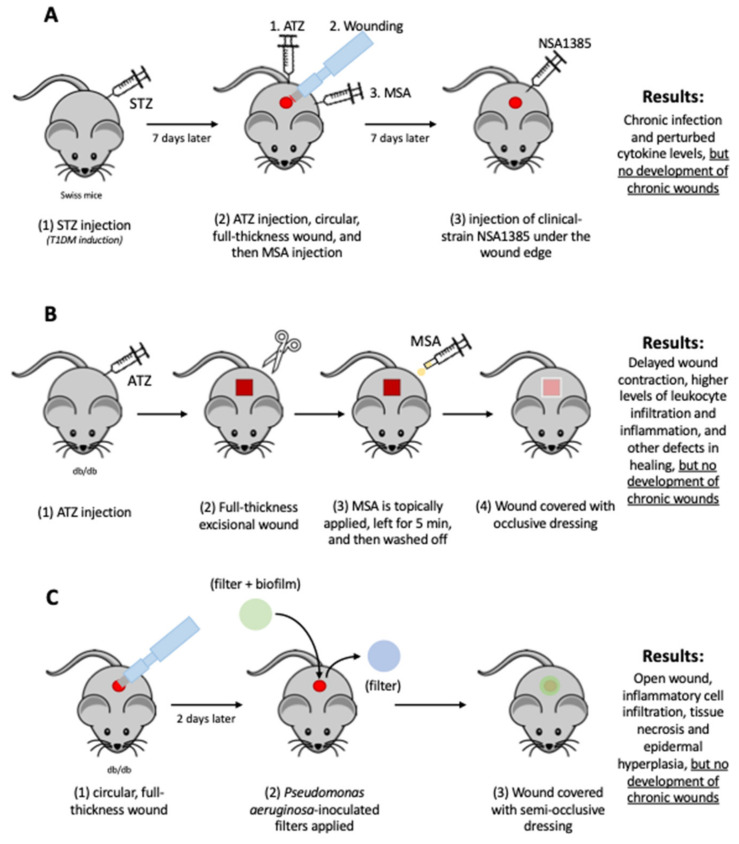
**Other chronic wound mouse models:** (**A**) Swiss mice are (1) administered STZ to induce T1DM. (2) Seven days later, mouse is injected with ATZ (catalase inhibitor), subsequently wounded, and finally injected with MSA (GPx inhibitor). (3) Clinical strain of *S. aureus* NSA1385 is inoculated under wound edge. Models presents chronic infection and perturbed cytokine levels but fail to resemble chronic wounds. (**B**) *db*/*db*^−/−^ mice (1) receive ATZ injection, (2) are wounded, (3) are topically administered MSA, and (4) covered with an occlusive dressing. However, MSA is washed off 5 min later. Models show delayed wound contraction, higher levels of leukocyte infiltration and inflammation, and other defects in healing, but lack key characteristics of chronic wounds. (**C**) *db*/*db*^−/−^ mice are inflicted with (1) full-thickness excision wounds and, 48 h later, (2) inoculated with *P. aeruginosa* and (3) subsequently covered with dressing. Wounds remain open, and show infiltration of inflammatory cells, tissue necrosis, and epidermal hyperplasia. Yet, as with the previous two models, they do not demonstrate many key features present in human chronic wounds.

**Figure 4 ijms-25-03837-f004:**
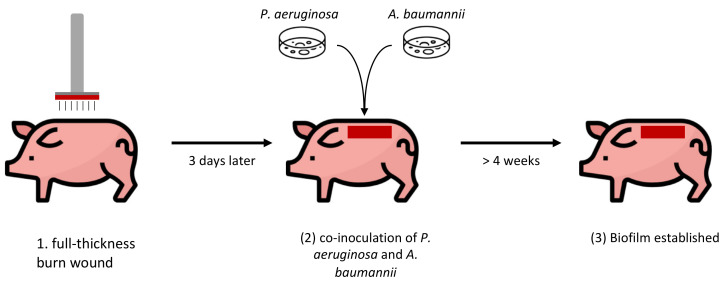
**Chronic wound model in pig:** (1) full-thickness thermal injuries are created on Yorkshire pigs via a microprocessor-controlled burning device. (2) Wounds are co-infected with *A. baumannii* and *P. aeruginosa* to recapitulate mixed-species biofilm infection as they appear in human chronic wounds. (3) Four weeks later, wounds reveal yellowish green discoloration and present established biofilm that compromises skin barrier function and is resistant to debridement.

**Table 1 ijms-25-03837-t001:** Chemically and genetically induced models for impaired wound healing studies.

Wound Type	Animal	Type of Model	Overview	Limitations	Use
Diabetic Impaired Healing	Murine and Porcine	Chemically induced	Diabetes mellitus is generated through the injection of streptozotocin before the creation of full-thickness excision wounds	STZ primarily generates type 1 diabetes mellitus Does not fully recapitulate symptoms of human diabetes and does not develop chronic wounds	Evaluating the efficacy of topically applied drugs, creams, nanomaterial, cells, adenovirus, and other molecules Investigating the role of specific genes in the healing process when created on knockout or transgenic mice
	Mice	Genetically modified	Full-thickness excision wounds are created on *ob*/*ob*, *db*/*db*, NONcNZO10/LtJ, and TallyHo/JnJ mice that exhibit characteristics of poor healing	The role of the leptin receptor is more significant in humans Does not fully recapitulate symptoms of human diabetes	Evaluating the efficacy of topically applied treatments

**Table 2 ijms-25-03837-t002:** Models for non-healing (chronic) wound studies in diabetes.

Wound Type	Animal	Type of Model	Overview	Limitations	Area of Study
Chronic Non-Healing	Murine	*db/db^−/−^ * mouse with elevated levels of oxidative stress	Full-thickness wounds are created on *db/db^−/−^ * mouse. Catalase and GPx are inhibited via ATZ and MSA, respectively, creating oxidative stress in the wound tissue.	The role of the leptin receptor is more significant in humans Does not fully recapitulate symptoms of human diabetes	Chronic wound initiation and progression and wound biofilm development Evaluating the efficacy of topically applied treatments including antibacterial drugs
	Porcine	Burn injury and bacterial inoculation	Full-thickness burn injuries are created on pigs followed by mixed-species bacterial inoculation.	No naturally developed biofilm Often used with burn wounds which involve a different healing process from excisional wounds	Understanding the host response to chronic infections and the effects of bacterial infection on wound healing
		Ischemic environment	Wound tissue is rendered ischemic via the creation of full-thickness bipedicle skin flaps; subsequently, full-thickness wounds are generated on the ischemic tissue.	Fails to naturally develop biofilm	Studying the biology of wound healing on ischemic tissue

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
