# Peer review of "Assessing Animal Models to Study Impaired and Chronic Wounds"

_ijms, 2024, doi:10.3390/ijms25073837_

Round 1
Reviewer 1 Report
Comments and Suggestions for Authors
Hi
I have reviewed this review article. Review article is written well but need following modifications before further processing;
1) In introduction section, authors must incorporate a paragraph highlighting advances in wound healing approaches too.
2) Authors must compare presented mice and pig models with respect to other already presented models in order to highlight significance of mice and pig models.
3) Thoroughly review entire manuscript for grammatical errors
4) Being a review article enhance introductions section and number references in terms of literature added. References are too less while considering it as review article
5) Highlight ethical concerns while dealing with these models
Thank you
Comments on the Quality of English LanguageAll comments have been loaded inn author's section
Author Response
1) In introduction section, authors must incorporate a paragraph highlighting advances in wound healing approaches too.
Response: This review is not about wound healing approaches—it is about models—and therefore, we don’t see the need for such an incorporation.
2) Authors must compare presented mice and pig models with respect to other already presented models in order to highlight significance of mice and pig models.
Response: We do not understand the reviewer's suggestion. This review already compares mice and pig models. What does the reviewer mean by “other already presented models”?
3) Thoroughly review entire manuscript for grammatical errors
Response: We have reviewed the manuscript for grammatical errors.
4) Being a review article enhance introductions section and number references in terms of literature added. References are too less while considering it as review article.
Response: We have increased the number of references.
5) Highlight ethical concerns while dealing with these models
Response: We do not understand why ethical concerns should be in this review as ethical concerns are addressed by the institution where the research is performed.
Reviewer 2 Report
Comments and Suggestions for Authors
The manuscript titled "Assessing models to study impaired and chronic wounds" presents a compelling review exploring various animal models utilized in the examination of chronic wounds. The work is highly valuable and holds significant promise for both scientists and clinicians. Nevertheless, I have a few minor comments that I believe could enhance the overall quality of the manuscript. Firstly, it is crucial to incorporate the term "animal models" into the title to more accurately reflect the scope of the review. Additionally, it is important to note that the work does not delve into in vitro models, and this information should be explicitly mentioned. Furthermore, I recommend including details on the effectiveness of streptozotocin (STZ) in inducing diabetes. This addition would contribute to a more comprehensive understanding of the methodologies discussed in the manuscript. In the introduction, it would be beneficial to acknowledge the existence of in vitro models, such as whole skin or conventional skin cell cultures. Upon incorporating these suggestions, I recommend accepting the manuscript.
Author Response
We thank the reviewer for the positive comments. Below we address their comments and requests.
- It is crucial to incorporate the term "animal models" into the title to more accurately reflect the scope of the review.
Response: Thank you for this suggestion. We have incorporated the term in the title of the review.
- It is important to note that the work does not delve into in vitro models, and this information should be explicitly mentioned.
Response: We have now made a brief explicit mention of in vitro models in the introduction.
- I recommend including details on the effectiveness of streptozotocin (STZ) in inducing diabetes. This addition would contribute to a more comprehensive understanding of the methodologies discussed in the manuscript.
Response: We have now addressed in the text of the manuscript STZ effectiveness and induced complications accompanying the treatment. Thank you for this suggestions.
- In the introduction, it would be beneficial to acknowledge the existence of in vitro models, such as whole skin or conventional skin cell cultures.
Response: Please see 2 above.